# Cytotoxic 4-Hydroxyprorocentrolide and Prorocentrolide C from Cultured Dinoflagellate *Prorocentrum lima* Induce Human Cancer Cell Death through Apoptosis and Cell Cycle Arrest

**DOI:** 10.3390/toxins12050304

**Published:** 2020-05-07

**Authors:** Seon Min Lee, Na-Hyun Kim, Eun Ju Jeong, Jung-Rae Rho

**Affiliations:** 1Gyeongnam Department of Environment & Toxicology, Korea Institute of Toxicology, 17 Jegok-gil, Munsan-eup 52834, Korea; smlee84@kitox.re.kr (S.M.L.); nhkim@kitox.re.kr (N.-H.K.); 2Department of Agronomy and Medicinal Plant Resources, Gyeongnam National University of Science and Technology, Jinju 52725, Korea; 3Department of Oceanography, Kunsan National University, Gunsan 54150, Korea

**Keywords:** *Prorocentrum lima*, prorocentrolide, cytotoxic, apoptosis, cell cycle arrest, cancer

## Abstract

Prorocentrolide and its analogs, the novel naturally derived antitumor agents, have recently been identified in the dinoflagellate *Prorocentrum lima*. In the current study, the underlying inhibitory mechanisms of 4-hydroxyprorocentrolide (**1**) and prorocentrolide C (**2**) on the proliferation of human carcinoma cells were determined. **1** and **2** arrested the cell cycle at the S phase in A549 cells and G2/M phase in HT-29 cells, leading to apoptotic cell death, as determined using fluorescence-activated cell sorting analysis with Annexin V/PI double staining. Apoptosis induced by these compounds was associated with alterations in the expression of cell cycle-regulating proteins (cyclin D1, cyclin E1, CDK2, and CDK4), as well as alterations in the levels of apoptosis-related proteins (PPAR, Bcl-2, Bcl-xl, and survivin). These findings provide new insights into the antitumor mechanisms of 4-hydroxyprorocentrolide and prorocentrolide C and a basis for future investigations assessing prorocentrolide analogs as prospective therapeutic drugs.

## 1. Introduction

Cancer is a leading cause of human death worldwide. In 2012, cancer resulted in 8.2 million deaths [1], and according to the American Cancer Society, in 2016 [2], cancer was the second most common cause of death in Europe and the United States. Despite several efforts to develop new drugs for cancer treatment, the incidence of cancer and the resulting mortality rate persist globally. Recently, a new trend has focused on the discovery and development of compounds derived from marine natural products, presenting an extensive range of structural diversity and excellent physiological activities. Notably, marine organisms produce these unique secondary metabolites to overcome harsh and competitive environmental conditions in the ocean.

Prorocentrum is a marine dinoflagellate that produces the main toxins responsible for diarrhetic shellfish poisoning (DSP), such as okadaic acid (OA) and dinophysistoxin-1 (DTX-1). Reportedly, the consumption of shellfish containing DSP toxins causes non-fatal gastrointestinal symptoms, including vomiting, abdominal pain, nausea, and diarrhea [3,4]. OA and DTXs are potent inhibitors of serine/threonine phosphatases 1 and 2A (PP2A). Accumulating evidence suggests that OA induces cancer cell death through apoptosis [5,6,7], DNA breaks and adduct formation [8,9], chromosomal non-disjunction [10], and cell cycle arrest [9]. Recently, structurally diverse compounds with excellent biological activity have been isolated from Prorocentrum species, and these toxins are under investigation as antitumor agents, owing to their pharmacological potential against various cancer cells.

Prorocentrolide is one of the marine biotoxins mainly produced by dinoflagellates, also known as the “fast-acting toxin” that was first reported in 1984 by Tindall and co-workers in a mouse bioassay [11]. In 1988, prorocentrolide-A was first isolated from *P. lima* [12]. Subsequently, prorocentrolide B, a new toxin isolated from *P. maculosum* inducing the characteristic “fast-acting” symptoms, was reported by Hu et al. in 1996 [13]. These toxins are macrocyclic compounds which possess imine and spiro-linked ether moieties, and the imino group present in the chemical structure is considered a commonly supposed pharmacophore [14]. However, the cytotoxic potential of prorocentrolides and their underlying mechanism remain elusive.

In the ongoing search for bioactive toxins from cultured *P. lima*, recently, 4-hydroxyprorocentrolide (**1**) and prorocentrolide C (**2**) were reported, with demonstrated cytotoxicity against HCT-116, Neuro-2a, and HepG2 cells [15] (Figure 1). However, whether these prorocentrolide analogs induce apoptosis remains unknown. Furthermore, their underlying mechanisms in cancer cell death need to be elucidated. In the present study, we demonstrated the inhibitory effects of **1** and **2** on cancer cell proliferation and explored the cell death induced by prorocentrolide analogs in A549 and HT-29 cells through apoptosis and G2/M phase arrest.

## 2. Results

### 2.1. 4-Hydroxyprorocentrolide and Prorocentrolide C Inhibit the Proliferation of A549 and HT-29 Cancer Cells

To investigate the effects of 4-hydroxyprorocentrolide and prorocentrolide C on cell growth, four human carcinoma cells, A549 (lung cancer), HepG2 (hepatocyte cancer), HT-29 (colon cancer), PC3 (prostate cancer) cells, were exposed to each compound at various concentrations (0.5~20 µM) for 24 h (Figure 2). Among these four cell lines, A549 and HT-29 cells were maximally sensitive to 4-hydroxyprorocentrolide and prorocentrolide C. The IC_50_ values of the two compounds were 17.8 μM and 14.6 μM for A549 cells and 9.9 μM and 10.5 μM for HT-29 cells, respectively. The colony formation analysis revealed that fewer colonies were formed after **1** or **2** treatment in A549 (Figure 3A) and HT-29 cells (Figure 3B), respectively. The reduced colony number following treatment with the test compounds was demonstrated in a concentration-dependent manner. In both cell lines, colony formation was significantly decreased following treatment with **2** than with **1**. Cell invasion is one of the main process of cancer metastasis. Therefore, we performed Transwell assays to assess the ability of carcinoma cells to cross the Transwell membrane barrier in the presence of **1** or **2**. As shown in Figure 3A,B, treatment with **1** or **2** inhibited the migratory activities of A549 and HT-29 cells in a concentration-dependent manner.

### 2.2. 4-Hydroxyprorocentrolide and Prorocentrolide C Induce S and G2/M Phase Arrest by Regulating Cell Cycle-Regulated Proteins

To determine whether **1** and **2** inhibit cancer cell proliferation through the induction of cell cycle arrest, we investigated the cell cycle stages following exposure to compounds **1** and **2** in A549 and HT29 cells. As shown in Figure 4, treatment with **1** and **2** resulted in the characteristic accumulation of cells in the S phase of A549 and G2/M phase of HT-29 cells, with a corresponding decrease in the G0/G1 phase. In A549 cells (Figure 4A), exposure to **2** resulted in the accumulation of cells in the S phase in a concentration-dependent manner. Cells in the S and G2/M phases were marginally increased by **1,** with no statistical significance. The effects of **1** and **2** on the G2/M arrest of the cell cycle was better illustrated in HT-29 colon cancer cells (Figure 4B). In both **1** and **2** treated cells, increased cells were observed in the G2/M phase in a concentration-dependent manner. It has been reported that cyclin/CDK complexes and checkpoint proteins are responsible for cell cycle progression. To confirm the effects of **1** and **2** on cell cycle arrest, the expression levels of cell cycle regulators were measured using Western blotting. As shown in Figure 5, the expressions of Cyclin D1, CDK4, Cyclin E1, and CDK2 were downregulated, and the expression of p21 was upregulated by **1** and **2** in A549 and HT-29 cells. Consistently with the cell cycle arrest results, the inhibition of these regulators, following treatment with the test compounds, was more significant in HT-29 colon cancer cells.

### 2.3. 4-Hydroxyprorocentrolide and Prorocentrolide C Induce Apoptosis in A549 and HT-29 Cancer Cells

To confirm the involvement of apoptosis in **1-** and **2**-induced inhibition of cell proliferation, Hoechst 33342 staining and flow cytometric analysis were performed. As shown in Figure 6, morphological changes (nuclear fragmentation, white arrows) were observed in A549 and HT-29 cells treated with **1** and **2**. The apoptotic and necrotic populations of A549 and HT29 cells were detected using flow cytometric analysis with Annexin V-FITC/PI staining. After 24 h of exposure to compounds **1** and **2**, the early apoptotic (Annexin V-positive/PI-negative) cell proportion was increased to some extent but not significant. At a concentration of 5 µM, cells in early apoptotic phase increased by 2.4- and 5.8-fold that if the untreated controls by **1** and **2** in A549 cells, and 3.29 and 1.48 times in HT-29 cells, respectively. On the other hand, the late (Annexin V-positive/PI-positive) apoptotic cell populations increased with the increasing **1** and **2** concentrations (0, 1, 5, 10 µM) in both cell lines. Treatment with **1** and **2** (at a concentration of 10 μM) increased the late apoptotic or dead cell population by 10.21 and 32.00 times that of the untreated controls in A549 cells and 48.36 and 39.48 fold times that of the untreated controls in HT-29 cells, respectively.

### 2.4. 4-Hydroxyprorocentrolide and Prorocentrolide C Induce Caspase-Dependent Apoptosis in A549 and HT-29 Cancer Cells

Next, to confirm the ability of **1** and **2** to induce apoptotic cell death, Western blot analyses were conducted to investigate the expression of the downstream apoptotic proteins (Figure 7). After 24 h of **1** or **2** exposure, the protein levels of cleaved-PARP increased, whereas the expression of Bcl-2, Bcl-xl, and survivin decreased. Additionally, in the caspase activity assay, the activity of caspase-3/7 increased with increasing concentrations (0, 1, 5, 10 µM) of **1** or **2** in both A549 and HT-29 cells. It is well known that cleavage of PARP, caspase 3/7 activation, and reduced Bcl-2, Bcl-xl, and survivin expression indicate the development of apoptosis. The JC-1 fluorescent cationic probe is widely used as an indicator of mitochondrial membrane potential. In the mitochondria, the JC-1 dye demonstrates potential-dependent accumulation. As **1** and **2** induced apoptotic cell death, we aimed to verify whether these compounds affect mitochondrial membrane potential in A549 and HT-29 cells. As shown in Figure 7, JC-1 was reduced after treatment with **1** or **2,** suggesting that mitochondrial depolarization was induced in both A549 and HT-29 cells. Collectively, these data indicate that **1** and **2** inhibited A549 and HT-29 cancer cell growth by inducing apoptosis.

## 3. Discussion

Among the various marine dinoflagellates species, *P. lima* is abundant worldwide and is found in tropical to cold water regions [16]. *P. lima* is known to produce several toxins, mainly OA and dinophysistoxins (DTX1 and DTX2), which are responsible for incidents of DSP [17,18,19,20]. OA and its analogs are reportedly inhibitors with high specificity for serine/threonine protein phosphatases [21,22]. The toxic mode of action of OA has been well investigated in some cell types [8,23,24]. In addition to OA and DTX, the other bioactive compounds that have recently attracted attention are prorocentrolides, which are members of the cyclic imine toxins possessing the characteristic cyclic imine group. Of these compounds, prorocentrolide-A was the first discovered compound from the Prorocentrum species in 1988 [12], and to date, seven prorocentrolide analogs have been reported. Amar et al. (2008) have recently investigated the action of prorocentrolide-A on both muscle and neuronal nicotinic acetylcholine receptors (nAChRs) [25]. However, the biological activities of prorocentrolide analogs remain undetermined. Our research team has recently reported the cytotoxicities of 4-hydroxyprorocentrolide and prorocentrolide C, a new prorocentrolide analog derived from cultured *P. lima* dinoflagellates [15]. To the best of our knowledge, the mechanism of action of prorocentrolides as a toxin has not been previously investigated. In the present study, we focused on the poorly understood actions of 4-hydroxyprorocentrolide and prorocentrolide C in human carcinomas. We performed cell viability assays screening against generally used human carcinomas, A549, HepG2, HT-29, and PC3 cancer cells. 4-Hydroxyprorocentrolide and prorocentrolide C inhibited the proliferation of human A549 lung cancer and HT-29 colon cancer cells in a concentration-dependent manner.

Apoptosis is the main pathway of chemotherapy agent-induced cell death. Here, we revealed, for the first time, that 4-hydroxyprorocentrolide and prorocentrolide C increased the apoptotic cell population in A549 and HT-29 cells. Signaling through activation of highly specialized families such as cysteinyl-aspartate protease, caspase, is one of the most common and major signaling cascades involved in apoptosis. Activated caspase is known to initiate cell death by subsequently cleaving and activating several major proteins including PARP-1 [26]. Cleavage PART-1 by capase is considered a characteristic of apoptosis [27]. This prorocentrolides upregulated the expression of the anti-apoptotic proteins Bcl-2, Bcl-xl, and survivin, suggesting the involvement of caspase-dependent apoptosis. It is assumed that the loss of checkpoint control, which regulates normal passage through the cell cycle, is involved in cancer progression. Targeting the cell cycle is a novel approach in cancer therapy. The present study confirmed that 4-hydroxyprorocentrolide and prorocentrolide C induced arrest in the G2/M phase of the cell cycle in A549 and HT-29 cancer cells. Cell cycle progression is a series of processes that are tightly regulated and integrated by Cyclin/CDK complexes. In eukaryotic cells, mechanisms, including Cdc kinase and cyclin activation, and Cyclin-dependent kinase inhibition regulate cell cycle progression [28]. Reportedly, Cdc2/Cyclin B1 complexes involved in the G2/M phase arrest in cells subjected to oxidative stress [29,30]. In A549 and HT-29 cells, exposure to 4-hydroxyprorocentrolide and prorocentrolide C for 24 h induced a significant decrease in the expression of Cyclin B1 and Cdc2, suggesting that the Cdc2/Cyclin B1 pathway was involved in the S or G2/M phase arrest induced by these prorocentrolides. Cyclin D1, synthesized in the G1 phase, is the key regulator acting for the G0/G1 phase. Consistent with the results of the cell cycle distribution, expression of Cyclin D1 was attenuated in both cells following exposure to 4-hydroxyprorocentrolide and prorocentrolide C. Direct binding of p21 and Cdc2 inhibits Cyclin B1/Cdc2 activity, thereby inhibiting transition to the G2/M phase. When DNA damage occurs in human cells, p21, which acts as an inhibitor of CDK, plays an essential role in maintaining G2 arrest [29]. Treatment with 4-hydroxyprorocentrolide and prorocentrolide C increased the expression of p21 and decreased CDK2 and CDK4 expressions. Our data suggest that these prorocentrolides-induced G2/M arrests include mechanisms of p21 activation and inhibition of Cyclin D1 expression. Prorocentrolide C demonstrated a greater activity in both A549 and HT-29 cells than 4-hydroxyprorocentrolide. The structural difference between prorocentrolide C and the other previously reported prorocentrolides (4-hydroxy-, 30-sulfate-, 9,51-dihydro-, 14-O-acetyl-4-hydroxy-prorocentrolide, and prorocentrolide A) is that the ester bonded B ring is absent in prorocentrolide C [25]. Further research to determine how other prorocentrolides act as toxins in cancer cells will be valuable for developing novel anticancer drugs. In conclusion, our discovery demonstrates that 4-hydroxyprorocentrolide and prorocentrolide C inhibit human carcinoma proliferation through apoptosis and G2/M cell cycle arrest.

## 4. Materials and Methods

### 4.1. Materials

The Hoechst 33342, dimethyl sulfoxide, crystal violet, and monoclonal anti-β-actin antibody were purchased from Sigma-Aldrich (St, Louis, MO, USA). DMEM, RPMI 1640 medium, fetal bovine serum (FBS), penicillin/streptomycin, and phosphate-buffered saline (PBS) were purchased from Gibco Life Technologies (Grand Island, NY, USA). MUSE Annexin V & Dead Cell Kit and MUSE cell cycle kit were obtained from Millipore (Billerica, MAM USA). Primary antibodies, including Bcl-2 (#3498, 26 kDa), Bcl-xl (#2762, 30 kDa), survivn (#2808, 16 kDa), cleaved-poly (ADP-ribose) polymerase (PARP) (#9542, 89 and 116 kDa), CDK2 (#2546, 33 kDa), CDK4 (#12790, 30 kDa) cyclinE1 (#4179, 48 kDa), cyclinD3 (#2936, 31 kDa), cyclinB1 (#4135, 55 kDa), cdc2 (#9116, 34 kDa) were purchased from Cell Signaling Technology, Inc. (Beverly, MA, USA). The secondary antibody was purchased from Jackson ImmunoResearch Laboratories, Inc.

### 4.2. Cell Cultures

Human non-small cell lung cancer (A549), human hepatocellular cancer (HepG2), and human colon cancer (HT-29), and human prostate cancer (PC-3) cell lines were obtained from the American Type Culture Collection (Manassas, VA, USA). A549, HepG2, and PC-3 cells were cultured in DMEM (Gibco BRL, Grand Island, NY, USA) containing 10% (*v*/*v*) FBS and 1% (*v*/*v*) antibiotics (100 U/mL of penicillin and 100 μg/mL of streptomycin). The cell lines were maintained in an incubator at 37 °C under a 5% CO_2_ atmosphere. HT-29 cells were cultured in RPMI1640 (Gibco BRL, Grand Island, NY, USA) containing 10% (*v*/*v*) FBS and 1% (*v*/*v*) antibiotics (100 U/mL of penicillin and 100 μg/mL of streptomycin). The cell lines were maintained in an incubator at 37 °C under a 5% CO_2_ atmosphere.

### 4.3. Cell Viability Assay

Cell viability was measured using the Cell Counting Kit-8 (CCK-8), according to the manufacturer’s instructions. Briefly, the cells were seeded in 96-well plates (1 × 10^4^ cells/well) and cultured for 24 h. Then, the cells were treated with various concentrations of test compounds for 24 h. After incubation, 10 μL of CCK-8 solution was added to each well and incubated for 3 h at 37 °C. The absorbance was measured using a microplate reader (Bio-Tek Company, Winooski, VT, USA) at 450 nm. The experiments were performed in triplicate.

### 4.4. Colony Formation Assay

Cells were seeded at a density of 8 × 10^2^ per well in 6-well plates and cultured in a humidified incubator with 5% CO_2_ at 37 °C for 14 days until visible colonies were observed. Colonies were fixed with 4% paraformaldehyde and stained with crystal violet for 15 min at room temperature. The colonies were photographed and counted under the microscope.

### 4.5. Apoptosis Analysis

Cells were seeded in 6-well plates at a density of 1 × 10^6^ and then treated with the test compounds (0~10 μM) for 24 h. Cells were harvested and diluted with DMEM containing 1% FBS as a dilution buffer to a concentration of 5 × 10^5^ cells/mL. Next, 100 μL of Annexin V/dead reagent and 100 μL of the cell suspension were mixed in the dark for 20 min at room temperature. The cells were analyzed using the MUSE cell analyzer (Merck Millipore, Hayward, CA, USA).

### 4.6. Cell Cycle Analysis

Cells were seeded in 6-well plates at a density of 1 × 10^6^ cells and then treated with the test compounds (0~10 μM) for 24 h. Cells were harvested, washed twice with PBS, and fixed with 70% ice-cold ethanol at −20 °C for 3 h, treated with 200 μL of MUSE cell cycle reagent in each tube, and incubated for 30 min at room temperature in the dark. Cell cycle analysis was performed using a MUSE cell analyzer (Merck Millipore, Hayward, CA, USA). Two thousand cells in each group was counted for cell cycle analysis.

### 4.7. Morphological Apoptosis

Cells were seeded in 6-well plates at a density of 3 × 10^5^ cells and then treated with compounds to be tested (0~10 μM) for 24 h. Cells were washed with PBS, fixed with 4% paraformaldehyde for 30 min, and stained with Hoechst 33342 dye for 10 min at 37 °C. In stained cells, the morphological changes were assessed using a fluorescence microscope.

### 4.8. Mitochondrial Membrane Potential Assay

Mitochondrial membrane potential was analyzed using a JC-1 Δψm assay kit (Cayman Chemical Company, Ann Arbor, MI, USA), according to the manufacturer’s instructions. Cells were seeded in 6-well plates at a density of 3 × 10^5^ cells and then treated with the compounds to be tested (0~10 μM) for 24 h. The JC-1 staining solution was added into the culture medium and incubated for 30 min at 37 °C. Next, the cells were with PBS and the stained cells were analyzed by flow cytometry (FACS Calibur; BD Biosciences, San Jose, CA, USA).

### 4.9. Transwell Invasion Assay

Transwell assay was performed by using 24-well Transwell (Costar, Cambridge, MA, USA) membranes (polycarbonic membrane, diameter 6.5 mm, pore size 8 μm), with each cell coated with 10 μL Matrigel (matrigel:serum-free medium 1:8). After 4 h, the upper chambers (serum-free medium) were seeded with 100 μL of DMEM and medium cells (5 × 10^4^ cells per well), and the lower chambers were filled with 600 μL of DMEM with 10% FBS. Next, 100 μL of DMEM with the test compounds (0~10 μM) was added to the upper chambers. The invasive cells were fixed with 4% paraformaldehyde and stained with 0.05% crystal violet. The invasive cells were counted by light microscopy (200× magnification).

### 4.10. Caspase-3/7 Activity

The caspase-3/7 activity was measured in triplicates using the Caspase-Glo^®^ 3/7 assay kit (Promega, Madison, WI, USA). Cells were seeded in 96-well plates at a density of 1 × 10^4^ cells and then treated with the test compounds (0~10 μM) for 24 h. Caspase-3/7 activities were assessed according to the manufacturer’s instructions. Next, 100 μL of Caspase-Glo^®^ 3/7 Reagent was added to each well and incubated for 1 h at room temperature. Caspase activities were measured using a microplate reader (Biotek Instruments, Inc., Winooski, VT, USA).

### 4.11. Western Blot Analysis

Cells were seeded in 6-well plates at a density of 3 × 10^5^ cells and then treated with the test compounds (0~10 μM) for 24 h. Cells were washed with ice-PBS and extracted using RIPA containing a Protease Inhibitor Cocktail (Santa Cruz, CA, USA) for 40 min on ice. Protein lysates were centrifuged at 13,000× *g* for 30 min at 4 °C. The lysed protein (30 μg) was separated using sodium dodecyl sulfate polyacrylamide gel electrophoresis (8–12%) at 100 V and transferred to polyvinylidene difluoride membranes. The membranes were blocked with 5% nonfat milk in PBS with Tween^®^ (PBST) buffer for 1 h at room temperature. The membranes were incubated with primary antibodies at 4 °C overnight. Then, the membranes were washed three times with the PBST buffer and incubated with secondary antibodies for 1 h at room temperature. Bands were visualized using ECL (Thermo Scientific, Waltham, MA, USA) and calibrated using the Chemidoc Imaging System (Bio-Rad; Hercules, CA, USA). This density value of the protein bands was normalized to β-actin.

## Figures and Tables

**Figure 1 toxins-12-00304-f001:**
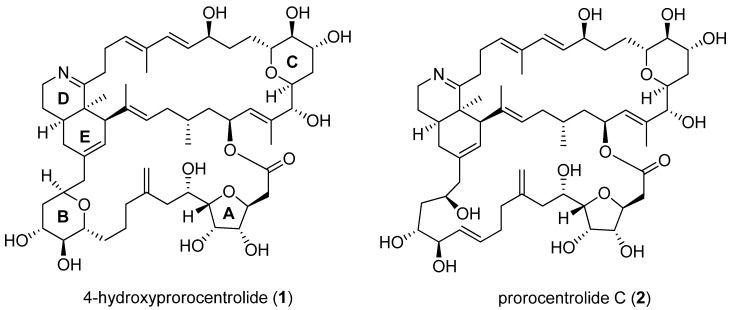
The structure of 4-hydroxyprorocentrolide (**1**) and prorocentrolide C (**2**) isolated from dinoflagellate *P. lima.*

**Figure 2 toxins-12-00304-f002:**
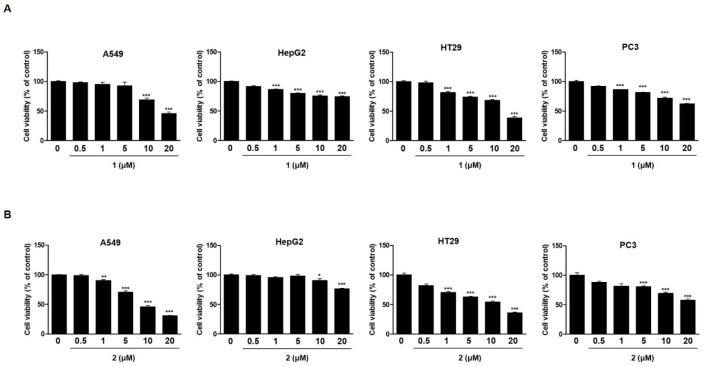
The antiproliferative effects of **1** and **2** on four human cancer cell lines, A549, HepG2, HT-29, and PC3, was determined by the MTT assay. Cells were treated with various concentrations (0.5, 1, 5, 10, 20 µM) of **1** (**A**) or **2** (**B**) for 24 h. The control group contained 0.01% dimethyl sulfoxide. Data represent the mean of five replicates. Each experiment was performed in triplicate; *** *p* < 0.001 compared to non-treated control cells.

**Figure 3 toxins-12-00304-f003:**
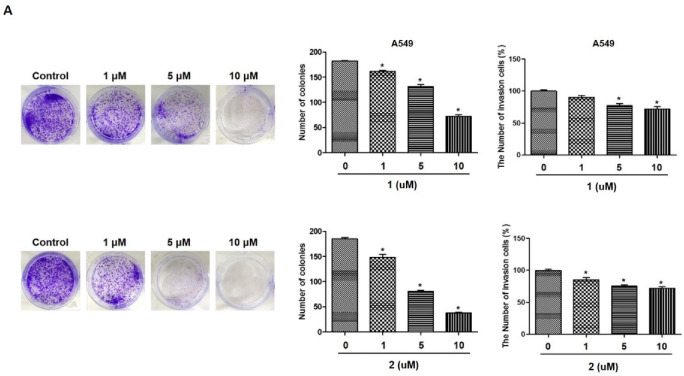
The effects of **1** and **2** on colony formation and Transwell invasion in A549 (**A**) and HT-29 (**B**) cells. For the colony formation assay, cells were treated with each compound at varying concentrations (1, 5, 10 µM) for 14 days until visible colonies were observed. For the Transwell invasion assay, the bottom chambers of Transwell plates were filled with 600 μL of Dulbecco’s Modified Eagle’s Medium (DMEM) containing various growth factors, whereas the top chamber was seeded with A549 or HT-29 cells in DMEM and treated with different concentrations (1, 5, 10 µM) of each compound for 24 h. The cells that migrated through the membrane were stained and counted. Results are presented as the mean ± standard deviation from three independent experiments; * *p* < 0.05 compared to non-treated control cells.

**Figure 4 toxins-12-00304-f004:**
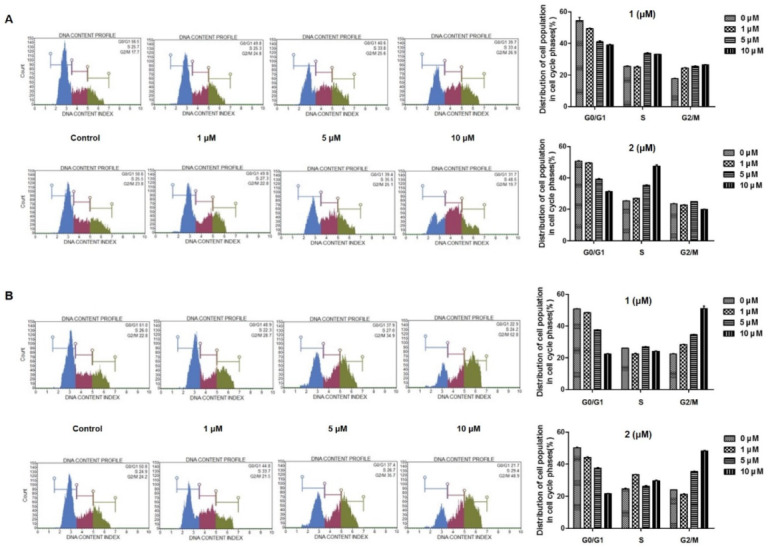
Effects of **1** and **2** on cell cycle arrest in A549 (**A**) and HT-29 (**B**) cells. Cells were treated with control or various concentrations (1, 5, 10 µM) of each compound for 24 h and analyzed by flow cytometry. The percentage of cell cycle distribution is presented as the mean ± standard deviation from three independent experiments.

**Figure 5 toxins-12-00304-f005:**
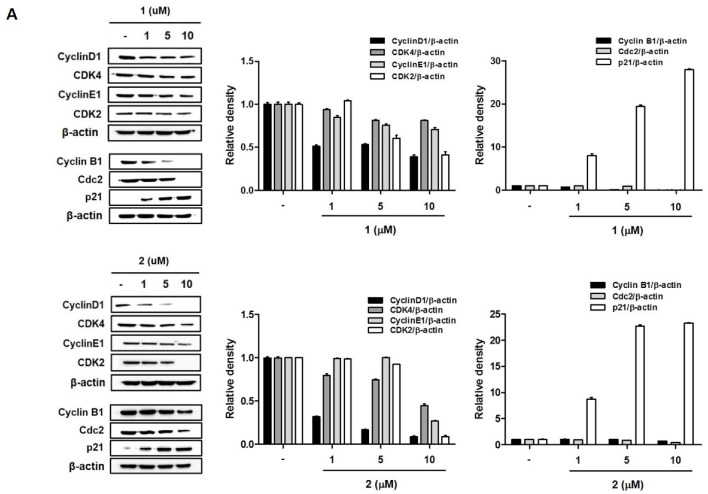
The effects of **1** and **2** on the expression of cell cycle-regulated proteins in A549 (**A**) and HT-29 (**B**) cells. Cells were treated with control or various concentrations (1, 5, 10 µM) of each compound for 24 h, and the protein levels of cyclin D1, CDK4, cyclin E1, and CDK2 were measured by Western blotting. Results are presented as the mean ± standard deviation from three independent experiments. The representative blots are presented.

**Figure 6 toxins-12-00304-f006:**
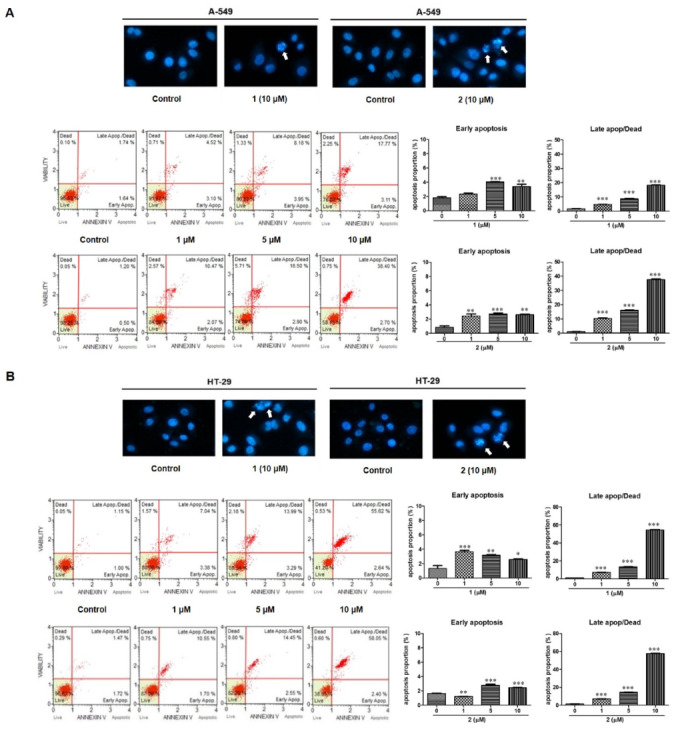
Apoptosis was induced by **1** and **2** in A549 (**A**) and HT-29 (**B**) cells. Apoptotic morphological changes were evaluated by fluorescent microscopy using Hoechst 33342 staining. White arrow indicates nuclear fragmentation/apoptotic bodies. Apoptotic cell proportion was analyzed using MUSE flow cytometry. Cells were treated with various concentrations (1, 5, 10 µM) of each compound and stained with Annexin V-FITC/PI according to the manufacturer’s instructions. The representative charts present the proportion of apoptosis. The percentage of cells in early and late apoptosis/dead phases is presented as the mean ± standard deviation from three independent experiments; * *p* < 0.05, ** *p* < 0.01 and *** *p* < 0.001 compared to non-treated control cells.

**Figure 7 toxins-12-00304-f007:**
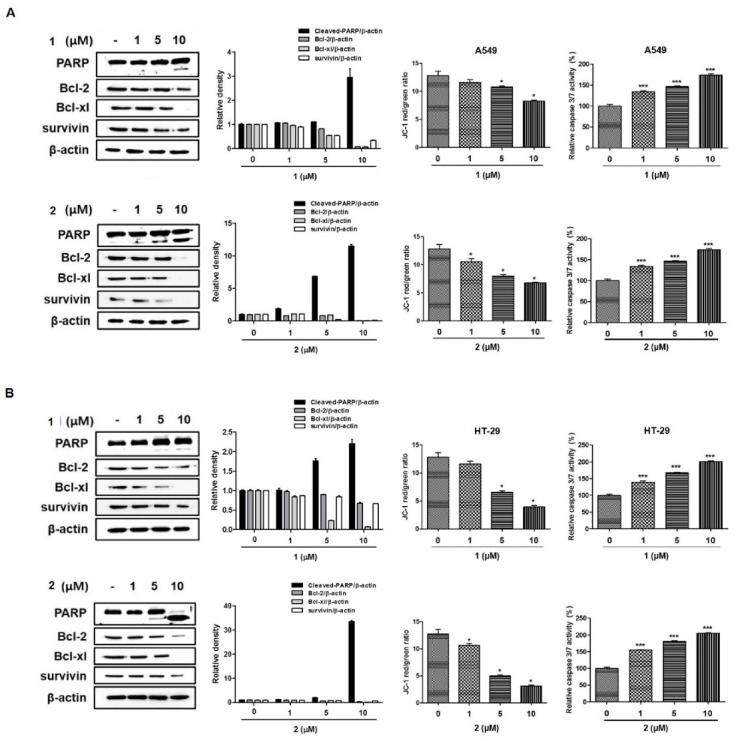
Compounds **1** and **2** induced caspase-dependent apoptosis and reduced mitochondrial membrane potential in A549 (**A**) and HT-29 (**B**) cells. Cells were treated with various concentrations (1, 5, 10 µM) of each compound for 24 h. The expressions of cleaved-PARP, Bcl-2, Bcl-xl, and survivin were determined by Western blotting, and caspase 3/7 activity was evaluated using the Caspase-Glo^®^ 3/7 assay kit. The mitochondrial membrane potential was measured with a JC-1 fluorescent probe and assessed by flow cytometry. The changes in the JC-1 red/green rate are presented as the mean ± standard deviation from three independent experiments. PARP, Poly (ADP-ribose) polymerase (PARP); Bcl-2, B-cell lymphoma 2; Bcl-xl, B-cell lymphoma extra-large. Results are presented as the mean ± standard deviation from three independent experiments; * *p* < 0.05 and *** *p* < 0.001 compared to non-treated control cells.

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
