# Peer review of "Cytotoxic 4-Hydroxyprorocentrolide and Prorocentrolide C from Cultured Dinoflagellate Prorocentrum lima Induce Human Cancer Cell Death through Apoptosis and Cell Cycle Arrest"

_toxins, 2020, doi:10.3390/toxins12050304_

Round 1

Reviewer 1 Report

The manuscript entitled “Cytotoxic 4-hydroxyprorocentrolide and prorocentrolide C from cultured dinoflagellate Prorocentrum lima induce human cancer cell death through apoptosis and G2/M cell cycle arrest” describes a series of studies aimed at defining the mechanism of cytotoxicity of two prorocentrolide analogs in various cancer cell lines.  An interesting and important study, the authors suggest that these prorocentrolide analogs results in growth arrest at the G2/M phase of the cell cycle.  Additionally, the authors suggest that apoptosis is the mode of cell death using a variety of assays including morphological assessment, externalization of phosphatidylserine, PARP cleavage, changes in the mitochondrial membrane potential, and caspase activity.  Overall, the data appears to generally support the authors conclusions, however some technical concerns in the analysis of the data exists calling into question the authors conclusions.  These points of concern are described below.

  1. The cell cycle data is not clear as the representative figure and quantification do not appear consistent (i.e. Figure 4A, compound 2 at 10 uM where an dramatic increase in S phase is observed in the figure, however the quantification shows a dramatic increase in G2/M phase. As this is a major point the study, the authors should consider using dot or contour plots to set initial gates for the cell cycle.  Furthermore, as the various phases of the cell cycle overlap to some degree, the authors should consider using a computer program such as ModFit to definitively define the boundaries between G1/ S phase and S/G2&M phase.
  2. In regards to the cell cycle data, was a doublet discrimination gate used on these plots? Additionally, where cells undergoing cell death included in the sample?  Significance of each phase should be noted, again as this is a major conclusion by the authors.
  3. Western blot data needs to be quantified over several independent experiments.
  4. In the version of the manuscript to review, the morphological changes with Hoechst is unclear.
  5. The authors need to show significance for the early apoptotic populations (Annexin V positive/ PI negative) to demonstrate apoptosis in figure 6. Conventionally, cell populations that are Annexin V positive and PI positive are considered late apoptotic or necrotic, therefore one cannot use this as a measure of apoptosis. 
  6. The authors state in the text (line 143) that the cleaved-PARP levels were determined after 48 hours, however the legend for figure 7 states 24 hours.
  7. As some of the most convincing data in regard to the cells undergoing apoptosis is the caspase assay, does inhibition of caspase activity (z-VAD or QVD-OPH) prevent cell death?

Author Response

Dear reviewer,

We thank you for providing the helpful and insightful comments to improve our manuscript. Our point-by-point response to your comments is provided below. The modified parts are marked in Red.

  1. The cell cycle data is not clear as the representative figure and quantification do not appear consistent (i.e. Figure 4A, compound 2 at 10 uM where an dramatic increase in S phase is observed in the figure, however the quantification shows a dramatic increase in G2/M phase. As this is a major point the study, the authors should consider using dot or contour plots to set initial gates for the cell cycle. Furthermore, as the various phases of the cell cycle overlap to some degree, the authors should consider using a computer program such as ModFit to definitively define the boundaries between G1/ S phase and S/G2&M phase. In regards to the cell cycle data, was a doublet discrimination gate used on these plots? Additionally, where cells undergoing cell death included in the sample? Significance of each phase should be noted, again as this is a major conclusion by the authors.

→ Thank you for your careful review on the manuscript. As pointed out, the wrong quantification graph for compound 2 was entered in Fig 4A. The cell population graph of compound 2 in Fig 4A was modified and the description was also corrected. We definitely agree with you that a computer program like ModFit can produce more accurate results, however, in the present study, we used MUSE cell analyzer for cell cycle analysis. While MUSE is easy to use, it has low compatibility to extract raw data and adjust data points using other programs. The DNA content profile in Figure 4 was automatically made in MUSE program.

  1. Western blot data needs to be quantified over several independent experiments.

→ As the reviewer’s comments, the quantification graph of western blot data in Fig. 5 and Fig. 7 were added. At least three independent experiments were performed to obtain western blot results.

  1. In the version of the manuscript to review, the morphological changes with Hoechst is unclear.

→ Thank you for your careful review on the manuscript. Result figure of Hoechst staining in Fig. 6 was enlarged and nuclear fragmentation/apoptotic bodies were indicated with arrows.

  1. The authors need to show significance for the early apoptotic populations (Annexin V positive/ PI negative) to demonstrate apoptosis in figure 6. Conventionally, cell populations that are Annexin V positive and PI positive are considered late apoptotic or necrotic, therefore one cannot use this as a measure of apoptosis.

→ Thank you for your comment. Annexin V positive and PI positive cells represent late apoptotic and necrotic cells. In many reports demonstrating apoptosis-mediated cell death, apoptotic proportion of cells was presented as the sum of early and late apoptotic cells (Nagappan et al., Oncology Letters, 2016, 12(2):1394-1402; Zhen et al., International Journal of Molecular Medicine, 2013, 589-596). In the present study, the increase of cells in early apoptotic phase was not significant by 1 or 2 treatment, while the increase in late apoptosis/dead phase was significant in high dose (10 uM). As the reviewer suggested, the graph in Fig 6 showing the percentage of apoptotic cells was separated; early apoptotic proportion and late apoptotis/dead proportion. Accordingly, the legend of figure 6 and text were re-described.

  1. The authors state in the text (line 143) that the cleaved-PARP levels were determined after 48 hours, however the legend for figure 7 states 24 hours.

→ Thank you for your careful review on the manuscript. The level of the cleaved-PARP was estimated after 24 hours. Accordingly, 48 h in line 174 was corrected to 24 h.

  1. As some of the most convincing data in regard to the cells undergoing apoptosis is the caspase assay, does inhibition of caspase activity (z-VAD or QVD-OPH) prevent cell death?

→ The cleavage of PARP through the activation of highly specialized families such as cysteinyl-aspartate protease, caspase, is known to be common and major signaling cascades involved in apoptosis. We observed that the exposure of A549 and HT-29 cells with compound 1 or 2 significantly increased the activity of caspase 3/7 and also induced the cleavage of PARP. In the present study, we attempted to find the signaling cascade involved in phylloketal derivative-mediated cell death for the first time. As the reviewer suggested, based on the newly obtained results in the present study, a more in-depth verification of compound-mediated apoptosis is planned to conduct in further study. The involvement of caspase and PARP cleavage is additionally described in Dicussion.

Reviewer 2 Report

The paper is original and brings novel, important data to the knowledge in the field. I recommend it to be published but after some corrections indicated below.

  • Fig.2- no information included in Fig.2 and Fig. 3 description about statistical differences
  • Fig.4- the lack of statistics neither in description of fig as well as on the graphs
  • The molecular weight of each analysed protein should be provided- Fig. 5 and Fig. 6
  • The accurate catalog number of antibodies used should be provided and the concentrations used. Also the species of antibodies

The western-blot results are crucial for estimating credibility of that research and obtained results therefore all these data should be included.

  • I’m surprised that as additional control the non-cancer human cells are not included, not subjected to any treatments. Sometimes apoptotic pathways are changed in cancer cells- the behavior of p53 can be an example. This control could also allow to better choose the effective and most optimal dose of the teasted reagent. And then densitometry would really help in that.
  • As for densitometry- I agree, your blots results are evident, especially for the 10uM but still the significant differences proved for 5uM would give more interesting points for discussion about involved pathways in apoptosis. In the long run, that could help in estimating the most effective and safe dose of the tested reagent. If you agree with me that it makes sense to do densitometry of the bands then Fig. 5 and Fig. 7 should be supplemented with the graphs of the densitometry analysis for each analysed protein. Statistics should be presented on the graphs. Densitometry analysis as well as the way of calculations should be described in methods section.
  • Line 126/124- Annexin V-positive/PI-positive is a marker of late apoptosis or necrosis? Please be precise.
  • Line 252- The cells were analyzed using the MUSE cell analyzer (Merck Millipore, USA)- describe how cells were evaluated according to the staining variants

AV+/PI+

AV-/PI+

AV+/PI-

AV-/PI-

How many cells were analysed?

  • Fig.6- Apoptotic morphological changes were evaluated by fluorescent microscopy using Hoechst 33342 staining.

What was observed as a result of this evaluation? Please name it and indicate by arrow on the photos.

Statistical differences are also not described in the description of the Fig 6.

  • Fig.7- Statistical differences are also not described in the description + The molecular weight of each analysed protein should be provided.
  • Why did you analysed only anti-apoptotic proteins? Some pro-apoptotic proteins should be analysed. It is especially important in terms of Bcl-2 family proteins analysis as the ratio of pro-apoptotic to anti-apoptotic proteins decides about the route the cell, death or live e.g. Bcl-2 to Bax; Bls xl/Bcl xs etc.
  • Caspase 3/7 activity and the changes in the JC-1 red/green rate – statistics is not described in Fig. 7 description.
  • “In conclusion, our discovery demonstrates that 4-hydroxyprorocentrolide and prorocentrolide C can inhibit human carcinoma proliferation by caspase-dependent apoptosis and G2/M cell cycle arrest”

Your results do not prove that caspase-dependent apoptosis took place in these experiments. There are two main apoptosis activation pathways: internal (mitochondrial) and external (related to death receptors). Caspase 3 and 7 are just executive caspases involved in both pathways in the last executive phase.

Author Response

Dear reviewer,

We thank you for providing the helpful and insightful comments to improve our manuscript. Our point-by-point response to your comments is provided below. The modified parts are marked in Red.

  1. Regarding to the lack of statistics

Fig.2- no information included in Fig.2 and Fig. 3 description about statistical differences.

Fig.4- the lack of statistics neither in description of fig as well as on the graphs. Statistical differences are also not described in the description of the Fig 6.

Fig.7 Statistical differences are also not described in the description.

Caspase 3/7 activity and the changes in the JC-1 red/green rate – statistics is not described in Fig. 7 description.

→ Thank you for your comment. Statistical differences are additionally described in the legend of Figure 2~7.

  1. The molecular weight of each analysed protein should be provided- Fig. 5 and Fig. 6. The accurate catalog number of antibodies used should be provided and the concentrations used. Also the species of antibodies. The western-blot results are crucial for estimating credibility of that research and obtained results therefore all these data should be included.

→ The molecular weight of each analysed protein and the catalog number of antibodies used in western blot is provided in method section “materials”.

  1. I’m surprised that as additional control, the non-cancer human cells are not included, not subjected to any treatments. Sometimes apoptotic pathways are changed in cancer cells- the behavior of p53 can be an example. This control could also allow to better choose the effective and most optimal dose of the teasted reagent. And then densitometry would really help in that.

Why did you analysed only anti-apoptotic proteins? Some pro-apoptotic proteins should be analysed. It is especially important in terms of Bcl-2 family proteins analysis as the ratio of pro-apoptotic to anti-apoptotic proteins decides about the route the cell, death or live e.g. Bcl-2 to Bax; Bls xl/Bcl xs etc.

→ In the present study, we attempted to find the signaling cascade involved in phylloketal derivative-mediated cell death for the first time. We agree with you that the effects of compounds 1 and 2 on non-cancer cells and pro-apoptotic proteins are also important in defining their action on Bcl-2 family proteins. However, in the very present time, it is impossible to perform the additional assays due to almost all the compound samples obtained from dinoflagellate Prorocentrum lima are exhausted. P. lima are in mass cultivation to obtain sufficient compounds. After that, a more in-depth verification of compound-mediated apoptosis is planned to conduct in further study.

  1. As for densitometry- I agree, your blots results are evident, especially for the 10uM but still the significant differences proved for 5uM would give more interesting points for discussion about involved pathways in apoptosis. In the long run, that could help in estimating the most effective and safe dose of the tested reagent. If you agree with me that it makes sense to do densitometry of the bands then Fig. 5 and Fig. 7 should be supplemented with the graphs of the densitometry analysis for each analysed protein. Statistics should be presented on the graphs. Densitometry analysis as well as the way of calculations should be described in methods section.

→ Thank you for your comment. As suggested, we performed densitometry experiments for the bands in Fig. 5 and 7. The quantification data was added in Fig. 5 and 7. Also, the method of densitometry analysis as additionally described in method section “western blot analysis”.

  1. Line 126/124- Annexin V-positive/PI-positive is a marker of late apoptosis or necrosis? Please be precise.

→ Thank you for your comment. Annexin V positive and PI positive cells represent late apoptotic and necrotic cells. In the present study, the increase of cells in early apoptotic phase was not significant by 1 or 2 treatment, while the increase in late apoptosis/dead phase was significant in high dose (10 uM). As the reviewer suggested, the graph in Fig 6 showing the percentage of apoptotic cells was separated; early apoptotic proportion and late apoptotis/dead proportion. Accordingly, the legend of figure 6 and text were re-described.

  1. Line 252- The cells were analyzed using the MUSE cell analyzer (Merck Millipore, USA)- describe how cells were evaluated according to the staining variants. How many cells were analysed?

→ Thank you for your comment. The detailed method of MUSE cell analysis including cell harvest, buffer volume, incubation temperature and time are provided in Methods. Cell number that is counted in analysis was added in method section, “4.5. Apoptosis Analysis and 4.6. Cell Cycle Analysis”.

  1. Fig.6- Apoptotic morphological changes were evaluated by fluorescent microscopy using Hoechst 33342 staining. What was observed as a result of this evaluation? Please name it and indicate by arrow on the photos.

→ Thank you for your careful review on the manuscript. Result figure of Hoechst staining in Fig. 6 was enlarged and nuclear fragmentation/apoptotic bodies were indicated with arrows.

  1. “In conclusion, our discovery demonstrates that 4-hydroxyprorocentrolide and prorocentrolide C can inhibit human carcinoma proliferation by caspase-dependent apoptosis and G2/M cell cycle arrest”. Your results do not prove that caspase-dependent apoptosis took place in these experiments. There are two main apoptosis activation pathways: internal (mitochondrial) and external (related to death receptors). Caspase 3 and 7 are just executive caspases involved in both pathways in the last executive phase.

→ Thank you for your comment. We corrected the conclusion to “our discovery demonstrates that 4-hydroxyprorocentrolide and prorocentrolide C inhibit human carcinoma proliferation through apoptosis and G2/M cell cycle arrest”. As the reviewer suggested, based on the newly obtained results in the present study, a more in-depth verification of prorocentrolide derivatives-mediated apoptosis is planned to conduct in further study.